# An Experimental Hut Evaluation of PBO-Based and Pyrethroid-Only Nets against the Malaria Vector *Anopheles funestus* Reveals a Loss of Bed Nets Efficacy Associated with *GSTe2* Metabolic Resistance

**DOI:** 10.3390/genes11020143

**Published:** 2020-01-29

**Authors:** Benjamin D. Menze, Mersimine F. Kouamo, Murielle J. Wondji, Williams Tchapga, Micareme Tchoupo, Michael O. Kusimo, Chouaibou S. Mouhamadou, Jacob M. Riveron, Charles S. Wondji

**Affiliations:** 1Vector Biology Department, Liverpool School of Tropical Medicine, Pembroke Place, Liverpool L3 5QA, UK; Murielle.Wondji@lstmed.ac.uk; 2Centre for Research in Infectious Diseases (CRID), LSTM Research Unit, Yaoundé 13591, Cameroon; mersimine.kouamo@crid-cam.net (M.F.K.); William.tchapga@crid-cam.net (W.T.); micareme.tchoupo@crid-cam.net (M.T.); gkusimo@gmail.com (M.O.K.); 3Faculty of Sciences, University of Yaoundé I, Yaoundé 337, Cameroon; 4Centre Suisse de Recherches scientifiques (CSRS), Yopougon 1303, Abidjan, Cote d’ivoire; mouhamadou.chouaibou@csrs.ci; 5Syngenta UK Limited, CPC4 Capital Park, Fulbourn, Cambridgeshire CB21 5XE, UK; Jacob.Riveron_Miranda@syngenta.com

**Keywords:** malaria, Long Lasting Insecticidal Nets, insecticide resistance, metabolic resistance, glutathione S-transferase, *Anopheles funestus*, piperonyl butoxide

## Abstract

Growing insecticide resistance in malaria vectors is threatening the effectiveness of insecticide-based interventions, including Long Lasting Insecticidal Nets (LLINs). However, the impact of metabolic resistance on the effectiveness of these tools remains poorly characterized. Using experimental hut trials and genotyping of a glutathione S-transferase resistance marker (L119F-*GSTe2*), we established that GST-mediated resistance is reducing the efficacy of LLINs against *Anopheles funestus*. Hut trials performed in Cameroon revealed that Piperonyl butoxide (PBO)-based nets induced a significantly higher mortality against pyrethroid resistant *An. funestus* than pyrethroid-only nets. Blood feeding rate and deterrence were significantly higher in all LLINs than control. Genotyping the L119F-*GSTe2* mutation revealed that, for permethrin-based nets, 119F-*GSTe2* resistant mosquitoes have a greater ability to blood feed than susceptible while the opposite effect is observed for deltamethrin-based nets. For Olyset Plus, a significant association with exophily was observed in resistant mosquitoes (OR = 11.7; *p* < 0.01). Furthermore, *GSTe2*-resistant mosquitoes (cone assays) significantly survived with PermaNet 2.0 (OR = 2.1; *p* < 0.01) and PermaNet 3.0 (side) (OR = 30.1; *p* < 0.001) but not for Olyset Plus. This study shows that the efficacy of PBO-based nets (e.g., blood feeding inhibition) against pyrethroid resistant malaria vectors could be impacted by other mechanisms including GST-mediated metabolic resistance not affected by the synergistic action of PBO. Mosaic LLINs incorporating a GST inhibitor (diethyl maleate) could help improve their efficacy in areas of GST-mediated resistance.

## 1. Introduction

The scale-up of insecticide-based interventions including Long Lasting Insecticidal Nets (LLINs) and Indoor Residual Spraying (IRS) has significantly contributed to the considerable reduction of malaria burden in the past decade [1]. Unfortunately, growing insecticide resistance in malaria vectors beside drugs resistance, lack of vaccine and reduced donor funding are threatening these successes. However, the actual impact of resistance, notably metabolic resistance, on the effectiveness of vector control tools against pyrethroid resistant mosquito populations remains a topic of debate especially when using entomological outcomes. Indeed, some studies have suggested that pyrethroid resistance does not yet impact the effectiveness of LLINs [2] whereas others have revealed a negative impact [3]. Furthermore, to help manage resistance, novel LLINs with the piperonyl butoxide (PBO), an insecticide synergist, have been designed by various manufacturers [4,5,6]. A randomized control trial comparing PBO-based nets to pyrethroid-only nets [3] led WHO to recommend in 2017 that national malaria control programmes and their partners should consider the deployment of pyrethroid-PBO nets in areas where the main malaria vector(s) have pyrethroid resistance that is: (a) confirmed, (b) of intermediate level, and (c) conferred (at least in part) by a monooxygenase-based resistance mechanism [7]. However, it remains unclear how these pyrethroid-PBO nets will performed when other mechanisms such as GST-based metabolic resistance are driving the resistance. One of the key challenges of assessing the impact of such metabolic resistance on the effectiveness of insecticide-based interventions such as LLINs has been the lack of molecular markers for resistance, as this phenotype is not easily associated with the outcome of the interventions. Recent efforts have detected a key genetic marker in the glutathione S-transferase epsilon 2 gene (*GSTe2*). This marker confer metabolic-mediated resistance to pyrethroids and dichlorodiphenyl-trichloroethane (DDT) in the major malaria vector *Anopheles funestus* in West and Central Africa [8]. Besides cytochrome P450s and esterases, GSTs are, one of the main enzyme families conferring metabolic resistance to insecticides [9] either through a direct metabolism or by catalyzing the secondary metabolism of substrates oxidized by cytochrome P450s [9]. Over-expression of GST epsilon 2 (*GSTe2*) has been associated with DDT and/or pyrethroids resistance in several mosquito species including *An. gambiae* [10], *An. funestus* [8] and *Aedes aegypti* [11]. In *An. funestus*, genomic and structural analyses revealed that a leucine to phenylalanine amino acid change (L119F) in *GSTe2*, has enlarged the substrate-binding pocket of the enzyme, conferring DDT/pyrethroid resistance in West and Central African populations [8]. This single amino acid change was used to design a simple field applicable DNA-based diagnostic tool [12] providing the opportunity to address questions regarding the direct impact of GST-mediated metabolic resistance on insecticide-based interventions such as LLINs.

The pyrethroid resistance in *An. funestus* across Africa is driven by metabolic resistance [13,14,15] as no knockdown resistance *(kdr)* has been reported so far for this species [16]. This predominance of the metabolic resistance mechanism in *An. funestus* through over-expression of detoxification genes such as GSTs or P450s makes this vector suitable to assess the impact of metabolic resistance on control interventions. The presence of the GST-mediated metabolic resistance in *An. funestus* populations such as in Cameroon [8,17,18] provides the opportunity to assess how the effectiveness of PBO-based nets is impacted when malaria vectors exhibit other type of metabolic resistance than cytochrome P450-based resistance which have been so far the only focus of synergists.

Experimental hut studies are the method of choice to evaluate the efficacy of LLINs against mosquito populations using entomological indices as a proxy for potential epidemiological impact [19,20]. It also provides relevant samples to assess how metabolic resistance impacts the effectiveness of LLINs as done previously to assess impact of target site mutations such as *kdr* [21].

Here, we assessed the performance of conventional pyrethroid-only nets versus PBO-based against a pyrethroid resistant *An. funestus* population from Cameroon using experimental huts. Furthermore, we took advantage of the L119F-*GSTe2* DNA-based diagnostic assay [8], to assess the impact of GST-mediated metabolic resistance on the performance of these five LLINs.

## 2. Materials and Methods

### 2.1. Study Area

The experimental hut station was located in Mibellon (6°4′60′′ N, 11°30′0′′ E), a village in Cameroon located in the Adamawa Region; Mayo Banyo Division and Bankim Sub-division. The Adamawa region is in the mountainous zone forming a transition between Cameroon’s forested south and savanna north. Malaria transmission is perennial with a high transmission as shown by very high infection rate of Plasmodium infection caused by *Plasmodium falciparum* but also *P. malariae* [12]. The village was located in close proximity to permanent water bodies including a lake and swamps which provide suitable breeding sites for *An. funestus* s.l. Human activities are mainly fishing, hunting and subsistence farming including maize, watermelon and coffee plantations. At the experimental station, 12 huts built following the World Health Organisation (WHO) standard [22], were available for a wide range of experimental hut trials. *An. funestus* s.s. was the main malaria vector in the area [18]. *Mansonia* sp., and *Culex* sp. were also present in the area. *Anopheles* mosquitoes in the area were highly resistant to pyrethroid and DDT [18]. The trial was carried out for 10 weeks during the rainy season between 10 July and 16 October 2016.

### 2.2. Experimental Hut Design

The huts are built following the prototype recommended by WHO for the West African region [22]. The hut is constructed on a concrete base surrounded by a drain channel to trap ants. The walls are made from concrete bricks and plastered inside and outside with a plaster made from a mixture of cement and sand. The roof is made from corrugated iron and the ceiling is made from plywood. The 4 windows located on three side of the hut are designed to create an angle with a 2 cm gap, which will facilitate the entry of mosquitoes flying upward and prevent the mosquitoes from escaping once they have entered the hut. A veranda trap is built at the back of the hut according to WHO protocol [23]. A curtain is used as a separation between the veranda and the rest of the hut. Before bedtime, each sleeper is required to raise the curtain to give mosquitoes the opportunity to take refuge in the veranda. In the morning, it is recommended that the sleeper lowers the curtain before starting the collection, to allow a separate collection of mosquitoes in the veranda and in the hut.

### 2.3. Net Treatment/Arm Comparison

During the experimental hut trial, four LLINs and one untreated net as negative control were compared (Table 1). These included two pyrethroid-only LLINs (PermaNet 2.0 and Olyset) and two PBO-based nets (PermaNet 3.0 and Olyset Plus) (Table 1). Each net was holed according to WHO protocol [23]. Six holes were generated (4 cm × 4 cm) per net, two on each of the long sides and one on each of the short sides.

#### 2.3.1. Hut Effect

Prior to the study, the hut effect was assessed to evaluate any specific attractiveness of huts. Untreated nets were hung in the six huts used for the study, and during 3 weeks, between the 10 and 29 May 2016, volunteers sleeping underneath collected mosquitoes each morning.

#### 2.3.2. Bioassays and Cone Assays

Cone bioassays were performed at the beginning of the study using the *An. coluzzii* susceptible Ngousso laboratory strain. This was done to confirm the quality of the five bed nets used in the study. Cone bioassays were also done in the insectary using F1 progeny from field collected *Anopheles funestus* from Mibellon. For PermaNet 2.0, PermaNet 3.0 (side and top), Olyset and Olyset Plus beside an untreated control net, five batches of 10 unfed females, 2–5 days old, were exposed to each bed nets using WHO cone assays for three minutes [22]. They were then transferred into a holding paper cup containers and the knock down was checked after 60 min and the mortality after 24 h post-exposure during which mosquitoes were provided sugar solution. Bioassays were also performed to generate highly resistant (alive after 90 min exposure) and highly susceptible (dead after 30 min) mosquitoes against 0.75% permethrin, 0.05% deltamethrin and 4% DDT WHO papers. The 30 and 90 min exposure tests were performed in separate tubes then scored 24 h post-exposure. For each test 4 replicates of 25 mosquitoes were used. The samples were then used to assess the association between L119F-*GSTe2* genotypes and resistance to these insecticides.

### 2.4. Experimental Hut Trial

The experimental hut trial was carried out during 60 night following the protocol described in the guidelines for laboratory and field-testing of long-lasting insecticidal nets [23]. To correct any specific attractiveness observed during the hut effect assessment, bed nets were rotated according to the Latin design square rotation [23] so that at the end of the study each net would have spent six days in each hut. The huts were cleaned weekly before the rotation of the nets. Six volunteer males were selected to sleep in the room from 20:00 GMT in the evening to 5:00 GMT in the morning. The sleepers were also rotated every day so that at the end of the week each sleeper would have spent one night in each hut. The rotation of the sleepers was done to correct any bias due to any specific attractiveness from the sleepers. In addition, the sleepers were blinded to the treatments.

#### 2.4.1. Mosquito Collection

Mosquitoes were collected every morning, using hemolysis tubes from: (i) inside the nets, (ii) in the room: floor, walls and roof, and (iii) in the veranda exit trap. Mosquitoes collected from each compartment were kept separately in a bag to avoid any mixing between samples from different compartments. Samples were then classified as dead, alive, blood fed or unfed. The ‘alive’ samples were kept in the paper cup and provided with sugar solution for 24 h and mortalities monitored.

The field-caught females were sorted according to morphological keys as previously described [24]. Mosquitoes belonging to the *An. funestus* group were species identified to species specific level using a cocktail PCR as previously described [25].

#### 2.4.2. Bed Nets Performance Assessment

The performance of the bed nets were expressed relative to control (untreated nets) in term of:Deterrence/entry rate: the reduction in hut entry relative to control. Deterrence (%) = 100 × (Du − Dt)/Du, where Du is the total number of mosquitoes found in untreated hut (control) and Dt is the total number of mosquitoes in the treated hut.Entry rate (%) = 100 × (Ht/Hn) where Ht is the total number of mosquitoes found in the hut and Hn is the total number of mosquitoes collected in all the 5 huts.Exophily (Excito-repellency): the proportion of mosquitoes found exited in the veranda trap Exophily (%) = 100 × (Ev/Et) where Ev is the total number of mosquitoes fund in veranda and Et is the total number of both inside the hut and veranda.Blood feeding rate (BFR). This rate was calculated as follows: Blood feeding rate = (N mosquitoes fed) × 100/total N mosquitoes. Where “N mosquitoes fed” was the number of mosquitoes fed, and “total N mosquitoes” was the total number of mosquitoes collected.Blood-feeding inhibition (BFI): the reduction in blood-feeding in comparison with the control hut. Blood feeding inhibition is an indicator of personal protection (PP). More precisely, the personal protection effect of each bed net is the reduction of blood feeding percentage induced by the net when compared to control. The protective effect of each bed net can be calculated as follows:Personal protection (%) = 100 × (Bu − Bt)/Bu, where Bu is the total number of blood-fed mosquitoes in the huts with untreated nets and Bt is the total number of blood-fed mosquitoes in the huts with treated nets [23].Immediate and delay mortality: the proportion of mosquitoes entering the hut that are found dead in the morning (immediate mortality) or after being caught alive and held for 24 h with access to sugar solution (delay mortality) [23]. In this study we focused on the overall mortality calculated as follows: Mortality (%) = 100 × (Mt/MT) where Mt is the total number of mosquitoes found dead in the hut and MT is the total number of mosquitoes collected in the hut.

#### 2.4.3. Ethical Clearance

The National Ethics Committee for Health Research of Cameroon approved the protocol of the study (ID:2016/03/725/CE/CNERSH/SP). Written, informed and signed consent was obtained from sleepers before starting the trials. The consent form provided all the information and the evaluation process about the study. Information was translated in local language when needed. All the volunteers involved in the study were followed-up and treated when showing malaria symptoms. All methods were performed in accordance with the relevant guidelines and regulations.

### 2.5. Impact of the L119F-GSTe2 Mutation on Insecticide-Treated Nets

The samples collected during the investigation on the performance of nets were grouped in several categories: dead, alive, blood fed, unfed; room and veranda and inside nets. The L119F-*GSTe2* mutation was genotyped in each group using an Allele Specific-PCR. This allows a direct measure of the relative survival and feeding success of resistant and susceptible insects in the presence of the different bed nets.

### 2.6. Genotyping

Samples classified as dead, alive, blood fed, unfed, room and veranda were used for DNA extraction using the Livak protocol [26]. The L119F-*GSTe2* mutation was genotyped to assess how the glutathione S-transferase gene, *GSTe2*, impacts the performance of the bed nets. An allele specific PCR [27,28] was used to detect the three genotypes of the L119F-*GSTe2* mutation (homozygote resistant:RR, heterozygote resistant: RS and homozygote susceptible: SS). The PCR was carried out using 10 mM of each primer and 1 µL of gDNA as template in 15 µL reaction containing 10× Kapa Taq buffer A, 0.2 mM dNTPs, 1.5 mM MgCl2, 1 U Kapa Taq (Kapa Biosystems, Wilmington, MA, USA). Amplification was carried out using thermocyclic parameters: 95 °C for 5 min; 30 cycles of 94 °C for 30 s, 58 °C for 30 s, 72 °C for 45 s, and final extension at 72 °C for 10 min. The following primers were used: L119F-Fwd: ATG ACC AAG CTA GTT CTG TAC ACG CT; L119F-Rev: TTC CTC CTT TTT ACG ATT TCG AAC T; L119F-Res1: CGG GAA TGT CCG ATT TTC CGT AGA AtAA; L119-F-Sus1: CAT TTC TTA TTC TCA TTT ACA GGA GCG TAaTC. PCR products were separated on 2% agarose gel by electrophoresis. The bands corresponding to different genotypes were interpreted as previously described [12,28].

### 2.7. Data Analysis

#### 2.7.1. Experimental Hut Trial

To calculate the proportion of each entomological outcomes and the level of significance between the treatments and between the control for each entomological outcomes, the XLSTAT software (Addinsoft, Berkeley, CA, USA) was used, as done previously [29,30]. The numbers of mosquitoes collected in the huts with different treatments were analysed by negative binomial regression. The effects of the treatments on each of the main proportional entomological outcomes (exophily, blood feeding and mortality) were assessed using binomial generalized linear mixed models (GLMMs) with a logit link function, fitted using the ‘lme4’ package for R 3.6. (R Development Core Team, 2019). A separate model was fitted for each outcome and for each mosquito species. In addition to the fixed effect of each treatment, each model included random effects to account for the following sources of variation: between the five huts used in the studies; between the five sleepers who slept in the huts; between the ten weeks of the trial.

#### 2.7.2. Test of Association between L119F-Mutation and the Entomological Outcomes

To investigate the association between the GSTe2 mutation and the ability of the mosquitoes to survive, blood feed or escape, Vassar stats was used to estimate the Odds ratio (OR) based on a fisher exact probability test with a 2 × 2 contingency table.

#### 2.7.3. Hut Effect Analysis

The one-way analysis of variance (ANOVA) using Prism 7.0 (GraphPad, San Diego, CA, USA) was used to determine whether there were any statistically significant differences between the means of the mosquitoes collected from the six huts. In this study, for all the analyses, an alpha of 0.05 was used as the cut off for significance.

## 3. Results

### 3.1. Cone Assays Using the An. gambiae Susceptible Lab Strain Ngousso

At the beginning of the study each bed net was exposed to the *An. coluzzii* susceptible lab strain Ngousso using WHO cone assays. Olyset plus and PermaNet 3.0 showed a mortality of 100 ± 00%, PermaNet 2.0 showed 93.3 ± 3.33% mortality and Olyset gave mortality rate of 96.97 ± 3.33% whereas no mortality was recorded for the untreated net (Figure 1A).

### 3.2. Cone Assays with An. Funestus from the Field (Mibellon)

Olyset and Olyset plus showed mortality of 5.63 ± 3.2%, and 67.23 ± 3.4% respectively when exposed to the F_1_ population of *An. funestus* from Mibellon using WHO cone assays. PermaNet 2.0 showed 25.06 ± 5.06% mortality and PermaNet 3.0 side and PermaNet 3.0 top gave 71.04 ± 3.33% and 100 ± 0% respectively whereas no mortality was recorded for the Yorkool and the untreated net (Figure 1A).

### 3.3. Hut Effect

After 18 days collections, a total of 1147 mosquitoes were collected in the huts with untreated net. Out of the 1147, 488 were females *An. funestus*, 428 *An. funestus* males, 195 *Mansonia* spp., 32 *Culex* sp., and four *An. rufipes*. The average of *An. funestus* females collected per room after 18 days ranged from 3 to 8 per day (Figure 1C). No significant difference (*p* = 0.09; DF = 5) between the numbers of mosquitoes collected in the different huts was observed (Figure 1C).

### 3.4. Mosquito Abundance

A total of 4656 mosquitoes were collected in five huts by human volunteers sleeping in the huts for 10 weeks corresponding to 360 man-nights. Out of the mosquitoes collected 1155 (25%) were *An. funestus.* s.s. females, 2421 (52%) were *An. funestus* males, 1004 (22%) were *Mansonia* spp., 03 (0.06%) were *An. gambiae* ss., 66 (1%) were *Culex* sp. and 04 (0.08%) were *Aedes* sp. (Figure 2A). However, only female *An. funestus* and *Mansonia* sp., due to its significant nuisance in this area, were considered for analysis (Table 2 and Table 3).

### 3.5. Performance of the Nets against An. Funestus s.s. Population

#### 3.5.1. Deterrent Effect/Entry Rate

In comparison to control, the entry rate was significantly reduced in all the huts for the five LLINs with the deterrence rates ranging from 39.2% for PermaNet 2.0 to 60.8% for PermaNet 3.0. PermaNet 3.0 had a significantly higher deterrence compared to PermaNet 2.0 but this was not the case for Olyset Plus (49%) over Olyset (54.9%) (Table 2).

#### 3.5.2. Induced Exophily Rate 

A low exophily rate was recorded in the control hut for *An. funestus* s.s. (6.9%). However, the exophily rate was significantly higher in the hut with Olyset (29.0%; *p* < 0.001), Olyset Plus (22.1%; *p* < 0.001), PermaNet 2.0 (17.7%; *p* < 0.001) and PermaNet 3.0 (17.0%; *p* < 0.001) compared to the control hut (Figure 2A and Table 2). No significant difference is observed between the five tested LLINs. A significant high exophily activity was observed during week 2 and week 9 (*p* < 0.05) (Appendix A).

#### 3.5.3. Mortality (Overall Mortality)

Low mortality rates were recorded in the control hut for *An. funestus* s.s. (5.4%). However, the mortality rate was significantly higher in the hut with Olyset Plus (25.1%; *p* < 0.01), PermaNet 2.0 (12.2%; *p* < 0.01) and PermaNet 3.0 (30.1%; *p* < 0.001) compared to the control hut. The mortality rate was significantly higher with the two PBO-based nets than all the pyrethroid-only nets (*p* < 0.001). However, no significant variation was observed between the mortality in the huts with Olyset (9.7 %; *p* ˃ 0.05) compared to control (Figure 2B and Table 2). It is also clear from our analysis that mortalities obtained were not influenced by the sleepers (*p* ˃ 0.05), by the huts (*p* ˃ 0.05) and by the weeks (*p* ˃ 0.05) (Appendix A).

#### 3.5.4. Blood Feeding Inhibition (BFI) 

High blood feeding rates were recorded in the control hut (40.8%). However, the blood feeding rate (BFR) was significantly lower in the hut with Olyset (BFR = 15.3; BFI = 62.3%; *p* < 0.001), Olyset Plus (BFR = 18.1%; BFI = 55.6%; *p* < 0.001), PermaNet 2.0 (BFR = 19.8%; BFI = 51.3%; *p* < 0.001) and PermaNet 3.0 (BFR = 15.7%; BFI = 61.5%; *p* < 0.001) compared to the control hut (Figure 2B and Table 2). PermaNet 3.0 had a higher BFI than PermaNet 2.0 but this was not the case with Olyset Plus compared to Olyset although the differences were not significant. We noticed a significant reduction of the blood feeding in hut 4 (*p* < 0.05) (Appendix A).

#### 3.5.5. Personal Protection (PP)

The lowest performance in terms of PP was recorded with PermaNet 2.0 (70.44%). PermaNet 3.0 provided a higher PP when compared to PermaNet 2.0 but not significant (84.9% vs. 70.4%; *p* ˃ 0.05). No significant difference was observed when comparing the PP provided by Olyset and Olyset Plus (83 vs. 77.3%; *p* ˃ 0.05) (Table 2).

### 3.6. Performance of the Nets against Mansonia spp. Population

#### 3.6.1. Deterrent Effect

In comparison to control, the entry rate was significantly reduced for the four LLINs with deterrence rates ranging from 36.8% for PermaNet 3.0 to 58.8% for Olyset. Contrary to *An. funestus*, PermaNet 2.0 had a higher deterrence rate (58.5%) compared to PermaNet 3.0 (36.8%) and similar for Olyset (58.8%) compared to Olyset Plus (52.3%) (Table 3).

#### 3.6.2. Induced Exophily Rate

A higher exophily rate was recorded in the control hut for *Mansonia* spp. (42.4%) than *An. funestus* (6.9%). However, the exophily rate of *Mansonia* spp., was significantly lower in the hut with Olyset (28.4%; *p* < 0.01), Olyset Plus (31.3%; *p* < 0.01) and PermaNet 3.0 (31.9%; *p* < 0.01) compared to the control hut. For PermaNet 2.0 no significant difference was observed when compared to control (37.3%; *p* ˃ 0.05) (Table 3).

#### 3.6.3. Mortality

The overall mortality rate recorded in the control hut for *Mansonia* spp. was 37.4%. However, this rate was significantly higher in the hut with Olyset Plus (68.1%; *p* < 0.001), PermaNet 2.0 (51.4%; *p* ˃ 0.05) and PermaNet 3.0 (65.3%; *p* < 0.001) compared to the control. No significant variation was observed between the mortality in the huts with Olyset (40.4%; *p* ˃ 0.05) compared to control (Table 3). The two PBO nets PermaNet 3.0 and Olyset Plus had a significantly higher mortality rate than the pyrethroid-only nets PermaNet 2.0 and Olyset (Table 3).

#### 3.6.4. Blood Feeding Inhibition (BFI)

The blood feeding rate recorded in the control hut for *Mansonia* spp. was 30.1%. However, this rate was significantly lower in the hut with Olyset (15.6%; BFI = 48.1%; *p* < 0.01), Olyset Plus (20.2%; 32.7%; *p* < 0.001), PermaNet 2.0 (16.2%; BFI = 46.2%; *p* < 0.01) and PermaNet 3.0 (9.3%; BFI = 69.26%; *p* < 0.001) compared to the control hut. PermaNet 3.0 had higher BFI than all the other nets but this was not significant when compared to PermaNet 2.0 and Olyset Plus.

#### 3.6.5. Personal Protection (PP)

The lowest PP was recorded with Olyset Plus (67.96%) whereas PermaNet 3.0 provided a higher PP but not significant when compared to PermaNet 2.0 (80.5% vs. 77.6%; *p* ˃ 0.05) but higher than the Olyset nets. No significant difference was observed when comparing the PP provided by Olyset and Olyset Plus (78.6% vs. 67.9%; *p* ˃ 0.05) (Table 3).

### 3.7. Comparative Analysis of the Impact of L119F-GSTe2 Mutation on the Efficacy of Conventional and PBO-Based Nets

#### Impact on Mortality

Due to the low number of dead mosquitoes (5) the impact on mortality could not be assessed for Olyset net. No correlation was observed between the L119F-*GSTe2* mutation and the ability to survive exposure to Olyset Plus (26 dead and 110 alive) when comparing the allelic frequency (OR = 0.85; *p* ˃ 0.05; CI 0.45–1.59). A similar result was observed for the genotypic frequencies including between RR vs. SS (OR = 0.61; *p* ˃ 0.05; CI 0.22–1.66), RS vs. SS (OR = 1.02; *p* = 1; CI 0.56–1.8) and RR vs. RS (OR = 0.59; *p* ˃ 0.05; CI 0.21–1.68) (Figure 3A and Table 4).

Similarly, no significant association was observed for PermaNet 2.0 between the mutation and the ability to survive (24 dead and 89 alive mosquitoes) when comparing the allelic frequency (OR = 1.18; *p* ˃ 0.05; CI 0.6–2.2) (Table 4) as well as the genotypic frequency of RS vs. SS (OR = 1.1; *p* = 1; CI 0.6–6.6) (Appendix A and Table 4).

For PermaNet 3.0, no significant association was observed between the mutation and the ability to survive when comparing the allelic frequency (OR = 1.42; *p* ˃ 0.05; CI 0.74–2.7) (Table 4) but comparing the genotypic frequency of RR vs. SS provided a higher Odds ratio close to significance (OR = 3.47; *p* ˃ 0.05; CI 1–11.9) (Appendix A).

### 3.8. Impact on Blood Feeding

For Olyset net, a significant association was observed between the 119F_*GSTe2* resistance allele and an increased ability to blood feed (21 blood fed and 92 unfed) when comparing the allelic frequencies (OR = 2; *p* < 0.05; CI 1.06–3.7) and the genotypic frequency of RS v SS (OR = 2.97; *p* < 0.001; CI 1.6–5.3) (Figure 3A and Table 5).

For Olyset Plus net, when considering the blood feed and unfed samples from the room only (23 blood fed and 49 unfed) a significant association was found between L119F-GSTe2 and an increased ability to blood feed when comparing the allelic frequencies (OR = 4.5; *p* < 0.001; CI 2.26–9.2). An even stronger association was observed when comparing the genotype frequencies between RR vs. SS (OR = 12.3; *p* < 0.001; CI 2.5–60.4) and RS vs. SS (OR = 8.42; *p* < 0.001; CI 4.37–16.2). But no association was established when comparing RR v RS (OR = 1.46; *p* ˃ 0.05; CI 0.29–7.2) (Figure 3B and Table 5). No association was also obtained when comparing RR v RS (OR = 1.07; *p* ˃ 0.05; CI 0.38–3.02) when genotyping all the mosquitoes fed and unfed collected in in the hut treated with Olyset (33 blood fed and 103 unfed).

For PermaNet 2.0, no association was observed when comparing the allelic frequency R vs. S (OR = 1.31; *p* ˃ 0.05; CI 0.55–2.01) (Figure 3C and Table 5). But an association was observed when comparing genotypic frequency RS vs. SS (OR = 1.4; *p* < 0.001; CI 0.77–2.53).

For PermaNet 3.0, a negative association was observed between the mutation and the ability to blood feed as resistant mosquitoes significantly blood fed less than susceptible when comparing the allelic frequency (OR = 0.35; *p* < 0.05; CI 0.17–0.73) and the genotypic frequency of RS vs. SS (OR = 0.5; *p* < 0.05; CI 0.26–0.92) (Figure 3D and Table 5).

### 3.9. Impact on Exophily

For Olyset, no association was observed between the L119F-*GSTe2* mutation and exophily (67 in room and 48 in veranda) when comparing the allelic frequency (OR = 1.17; *p* ˃ 0.05; CI 0.6–2.2) (Table 4) and the genotypic frequency RS vs. SS (OR = 1.18; *p* ˃ 0.05; CI 0.6–2.1) (Figure 4A and Table 6).

For Olyset Plus, an association was observed between the mutation and the ability to escape to the veranda when considering the allelic frequency (OR = 3.4; *p* < 0.001; CI 1.67–6.9) (Table 6). An even stronger association was observed between the mutation and a preference for the veranda when comparing RR vs. SS (OR = 11.76; *p* < 0.01; CI 2.59–53.4), RS vs. SS (OR = 2.99; *p* < 0.01; CI 1.58–5.65). But no significant association was observed between RR vs. RS (OR = 3.9; *p* = 1; CI 0.8–18.59) (Figure 4B,C and Table 6). For PermaNet 2.0, an association was observed between the L119F-*GSTe2* mutation and the ability to exit the room when comparing the genotypic frequency RS vs. SS (OR = 1.35; *p* < 0.01; CI 0.75–2.3). A stronger association was observed when assessing the impact only among the unfed mosquitoes for RS vs. SS (OR = 3.37; *p* < 0.001; CI 01.84–6.17) but not at the allelic level (Figure 4D and Table 6). For PermaNet 3.0, no association was observed between the L119F-*GSTe2* mutation and a preference for the room or the veranda when comparing the allelic frequency (OR = 0.94; *p* = 1; CI 0.5–1.8) (Table 6). The same trend was observed when comparing the genotypic frequency RR vs. SS (OR = 1.22; *p* = 1; CI 0.46–3.2), RS vs. SS (OR = 0.59; *p* ˃ 0.05; CI 0.31–1.12) RR vs. RS (OR = 1.68; *p* ˃ 0.05; CI 0.6–4.7) (Figure 4E and Table 6).

### 3.10. Correlation between L119F-GSTe2 and Mortality from Cone Assays

Due to the low number of dead mosquitoes obtained from the experimental huts, the impact of the L119F-*GSTe2* on the ability of mosquitoes to survive exposure to various nets was assessed using samples from cone assays. Only a few dead were obtained for Yorkool and Olyset, preventing the assessment of these nets.

#### 3.10.1. Olyset Plus

No association was observed between the mutation and the mortality (40 dead vs. 40 alive mosquitoes) at both allelic level (OR = 1.04; *p* = 1; CI 0.5–1.8) and genotypic for RR vs. SS (OR = 1.1; *p* ˃ 0.05; CI 0.4–2.7) (Figure 5A and Table 7).

#### 3.10.2. PermaNet 2.0

Resistant 119F-*GSTe2* mosquitoes exhibited a greater ability to survive than susceptible mosquitoes (34 dead and 46 alive) at both allelic level (OR = 1.8; *p* < 0.01; CI 0.9–3.5) and genotypic frequency RR v SS (OR = 2.09; *p* < 0.01; CI 1.1–4.2) (Figure 5B and Table 7).

#### 3.10.3. PermaNet 3.0

Resistant 119F-*GSTe2* mosquitoes when exposed to PermaNet 3.0 showed a greater ability to survive than susceptible mosquitoes even than with PermaNet 2.0 at allelic level (OR = 3.8; *p* < 0.001; CI 1.8–7.7). A higher correlation was further observed when comparing the genotypic frequencies for RR vs. SS (OR = 30.1; *p* < 0.001; CI 3.8–234), RS vs. SS (OR = 2.4; *p* <0.01; CI 1.3–4.5) and RR vs. RS (OR = 12.3; *p* < 0.001; CI 1.8–7.7) (Figure 5C and Table 7). This result applies only to PermaNet 3.0 side net as no mosquito survived exposure to PermaNet top containing PBO.

### 3.11. Validation of the Association between L119F-GSTe2 and Resistance to Pyrethroids and DDT in Mibellon

To further validate the impact of L119F-*GSTe2* on LLINs, we established the association between this marker and the ability to survive exposure to the pyrethroids used for Olyset (permethrin) and PermaNet (deltamethrin) nets besides DDT. Correlation analysis between L119F genotypes and deltamethrin revealed that homozygote resistant mosquitoes were significantly more likely to survive exposure to deltamethrin than both homozygote susceptible SS (OR = 4.6; *p* < 0.001) and heterozygotes (OR = 3.75; *p* < 0.001). However, there was no significant difference between RS and SS (Figure 5D). A similar pattern was observed for permethrin (Figure 5E and Table 5). Analysis of the correlation with DDT revealed a much stronger association between L119F and DDT resistance as RR exhibited a greater ability to survive exposure to DDT than SS (OR = 66.7; *p* < 0.001) and also more than RS (OR = 4.8; *p* < 0.001). Contrary to both pyrethroids heterozygote RS mosquitoes are also more able to survive exposure to DDT than homozygote susceptible (OR = 14.0; *p* < 0.001) (Figure 5F and Table 7).

## 4. Discussion

This study has taken advantage of the predominant presence of metabolic resistance in *An. funestus* and the recent availability of a simple DNA-based diagnostic tool for *GSTe2*, to establish the impact of GST-mediated metabolic resistance on the effectiveness of both pyrethroid-only and PBO-based LLINs against *An. funestus*.

### 4.1. PBO plus pyrethroid-Based Nets are More Effective than Pyrethroid-Only Nets against Pyrethroid Resistant An. funestus

#### 4.1.1. Mortality

The mortality rates obtained in this study with experimental huts were generally low (7.3 to 30.1%), but such mortality levels are similar to those observed in other experimental hut studies including in Benin [5], or recently in Burkina Faso where the mortality rates ranged from 9.5% to 46.1% in two locations of high pyrethroid resistance in *An. gambiae* [31]. However, higher mortality has been observed for another highly resistant pyrethroid resistant population of *An. gambiae* in Ivory Coast (Yaokoffikro) for PermaNet 3.0 (54%) and PermaNet 2.0 (34.7%) [32] than seen here for the *An. funestus* population in Mibellon, which could potentially be associated with differences in resistance mechanisms. The low level of mortality observed is likely due to the level of pyrethroid resistance in *An. funestus* in Mibellon with permethrin mortality around 48.88 ± 5.76% and deltamethrin mortality around 38.34 ± 5.79% [18]. Nevertheless, despite the ongoing resistance both PBO-based nets induced a significantly higher mortality than the pyrethroid-only LLINs against both *An. funestus* and *Mansonia* spp. This higher mortality of PermaNet 3.0 and Olyset Plus with *An. funestus* is also in line with the cone assay results although PermaNet 3.0 (Top) presented a higher mortality for both huts and cone assays than Olyset Plus as also seen for *An. gambiae* in Burkina Faso [31]. Overall, the higher mortality with both PBO-nets suggests that, in areas of pyrethroid resistance, these new generation LLINs (PermaNet 3.0 and Olyset Plus) provide a better protection against both malaria vectors and nuisance mosquitoes such as *Mansonia* spp.

#### 4.1.2. Blood Feeding

The blood feeding rate of all the five nets, an indicator of the personal protection, provided by the net was significantly lower (≤23.8%; *p* < 0.001) compared with the untreated control (40.8%). However, the blood feeding inhibition rate was higher for PermaNet 3.0 (61.5%) than PermaNet 2.0 (51.4%) this was not the case between Olyset Plus (55.6%) and Olyset (62.4%). Similar results have been obtained for *An. gambiae* in Benin [6] where comparable blood feeding inhibition rates were observed for Olyset Plus (85%) and Olyset (82%) despite the greater mortality rate with Olyset Plus (81%) vs. 42% for Olyset. The same pattern of blood feeding inhibition was observed here for *An. funestus* and *Mansonia* spp. with PermaNet 3.0 providing the highest inhibition level in *Mansonia* spp. whereas Olyset Plus was even lower than Olyset.

#### 4.1.3. Exophily

The excito-repellency effect of all the nets was at least twice higher compared to control showing that *An. funestus* mosquitoes were affected by the repellent effect of pyrethroids. These results are in line with previous results demonstrating the repellent effect of pyrethroid nets against *Anopheles* mosquitoes [19,32]. No significant difference was observed in the level of exophily between the conventional nets compared to synergist suggesting that PBO did not impact the repellent effect of pyrethroids. Comparing *An. funestus* to *Mansonia* spp., revealed a significant difference in term of exophily with only 6.9% of *An. funestus* exiting the hut with untreated net whereas 42.4% *Mansonia* spp. exited the room with control net. This is different from the previous observation in Ivory Coast where a similar exophily (~35%) was observed for both *An. gambiae* and *Mansonia* spp. in experimental huts [32]. Furthermore, while all the 5 LLINs significantly increased the exophily rate for *An. funestus*, this was not the case in *Mansonia* spp. with no significant change observed probably as *Mansonia* spp. are more exophilic [33].

### 4.2. GSTe2 Mediated Metabolic Resistance Is Reducing Efficacy of LLINs

With the design of the L119F*-GSTe2* diagnostic tool [8], this study has investigated for the first time the impact of GST-mediated metabolic resistance on the efficacy of LLINs. 

#### 4.2.1. Experimental Hut Study

A significant association was observed between the blood feeding ability of *An. funestus* and the L119F*-GSTe2* mutation as mosquitoes with the 119F resistance allele have significantly higher blood feeding rate compared to those with L119 susceptible allele against Olyset Net (*p* < 0.001) and Olyset plus (*p* < 0.001). This is similar with observations for the cytochrome P450s *CYP6P9a* and *CYP6P9b* for which the resistant alleles *CYP6P9a_R* and *CYP6P9b_R* were recently shown, in a release-recapture experimental hut study, to also provide a greater ability to blood feed and to survive exposure to pyrethroid-only LLIN (PermaNet 2.0) [14,34]. This suggests that L119F-*GSTe2* mutation likely contributes to an increased malaria transmission as every additional bite increases the chance of sporozoite to be passed to the populations. This is particularly a concern as 119F-RR resistant mosquitoes have also been shown to live longer [27,28] and to have a greater vectorial capacity to transmit *Plasmodium* [12]. However, it is noticeable that the impact of L119F*-GSTe2* on blood feeding ability is mainly seen for Olyset Plus and Olyset LLINs impregnated with permethrin but present an opposite effect for PermaNet 3.0 and 2.0 both impregnated with deltamethrin. The cause of such difference between permethrin- and deltamethrin-based nets is unclear at this point since the greater blood feeding in Olyset net is not associated with greater survivorship. Functional analyses with transgenic Drosophila, In vitro metabolism assays with recombinant *GSTe2* enzyme combined with genotype/phenotypes analyses had shown that *GSTe2* was able to confer resistance to both permethrin and deltamethrin but more so to permethrin [35,36]. This could partly explain the differences observed between permethrin- and deltamethrin-based nets although further studies will be needed to fully establish the underlying reason. Similar genotype/phenotype studies performed to assess the impact of the knockdown target-site resistance (*kdr*) on resistance did not reveal such a significant association with blood feeding potentially due to the fact that *kdr* frequency was already very high in tested *An. gambiae* population [21]. However, indirect comparison of *kdr* effect in two separate populations of *An. gambiae* in Benin; one with high *kdr* frequency (Ladji) and another with low frequency (Malanville) did suggest that *kdr* could also impact the efficacy of LLINs [19].

Interestingly, the L119F-GSTe2 mutation was also associated with an increased exophily for Olyset Plus and PermaNet 2.0 suggesting that the increased expression of GSTe2 could impact mosquito’s behavior helping them avoids exposure to insecticides. Similarly, over-expression of carboxylesterase in *Culex pipiens* [37] has been previously associated with a behavioral change as resistant mosquitoes were found to have a reduced mobility [37] supporting that metabolic resistance could impact mosquitoes’ behavior in the presence of insecticide-based interventions.

No significant association was observed between L119F-*GSTe2* and mortality in the experimental hut’s samples for all LLINs potentially due to the low number of dead mosquitoes which did not provide statistical power coupled with lower frequency of RR in this location. However, a trend was observed for PermaNet 3.0 with an increased proportion of 119F homozygote resistant mosquitoes able to survive exposure to this LLIN than both homozygote susceptible L119-SS (OR = 3.47; *p* ˃ 0.05) and heterozygotes (OR = 3.46; *p* ˃ 0.05).

However, other mechanisms than GSTs are also at play in this location of Mibellon, as shown recently [14], revealing the over-expression of cytochrome P450 genes such as CYP6P5 and CYP325a. Unfortunately, there is not yet a molecular marker to help assess the impact of these genes with the different samples collected after experimental huts. Generating more DNA-based markers as done for the CYP6P9a/b [14] and for L119F-GSTe2 [8] will further help to gain a more complete picture of the impact of resistance as a whole. No kdr target-site resistance has yet been detected in An. funestus (Irving et Wondji 2017) including in Mibellon [18].

#### 4.2.2. Impact from Cone Assays

The trend for a correlation between *GSTe2* and reduced mortality with LLINs was supported by results from cone assays where a very strong association was observed between L119F-*GSTe2* and ability to survive exposure to PermaNet 3.0 (side) (OR = 30.1) and PermaNet 2.0 although at lower extent (OR = 2.09). The lack of association observed with Olyset Plus could be explained by the effect of PBO which for this net is incorporated on the entire net whereas for PermaNet 3.0 PBO is only on the top. The PermaNet 3.0 (side) used here does not contain PBO. One reason why a greater association is found with PermaNet 3.0 (side) could be the fact that with higher deltamethrin concentration than PermaNet 2.0, mosquitoes surviving exposure to PermaNet 3.0 are those likely to be much more resistant because of a greater expression of GSTe2 and other cytochrome P450s. So the higher dose of deltamethrin in PermaNet 3.0 has further selected mosquitoes with 119F-*GSTe2* resistant allele than with the lower dose in PermaNet 2.0. This is also the likely reason why a trend of a greater ability to survive was seen in the experimental hut for PermaNet 3.0.This suggests that mosquitoes displaying several resistance mechanisms than just P450-based resistance could avoid even PBO-based nets as when cytochrome P450 enzymes are inhibited by PBO, mosquitoes bearing other mechanisms, as in this case the 119F resistance allele, are able to better survive exposure to PBO-based nets. However, this cannot explain why such results were not observed with Olyset Plus the other PBO-net. However, the spread of a *GSTe2*-mediated metabolic resistance conferring the ability of mosquitoes to survive exposure to synergist LLINs will constitute a concern at a time when such nets are gradually being introduced to fight pyrethroid resistant mosquitoes. GST resistance will also need to be taken into consideration than just cytochrome P450 resistance. Nevertheless the greater efficacy seen with the roof of PermaNet 3.0 in cone bioassays shows that P450 role remains important. Future studies will need to also perform synergist assays with DEM and PBO to establish the contribution of both GSTs and cytochrome P450s to better inform of the likely efficacy of LLINs.

## 5. Conclusions

This study has revealed a loss of efficacy of LLINs against pyrethroid resistant populations of *An. funestus* in Cameroon using experimental huts. However, PBO based nets (PermaNet 3.0 and Olyset Plus) are more efficient than conventional pyrethroid-only nets. It is the reason why as suggested by WHO 2017 report we recommend the deployment of PBO nets in area where pyrethroid resistance is confirmed and at least partly conferred by P450. Noticeably, the GST-mediated metabolic resistance not impacted by PBO could threaten the continued effectiveness of the PBO-based LLINs and the impact of this mechanism needs also to be considered to maximize the effectiveness of LLINs. One option could be to generate new mosaic LLINs also incorporating the Diethyl Maleate (DEM) beside PBO to inhibit GST metabolic activity and further increase the efficacy of these nets. However, further studies will need to be performed before including establishing the extent of synergism provided by DEM in various mosquito populations and the most suitable position on the net (either on top as PBO or on the entire net).

## Figures and Tables

**Figure 1 genes-11-00143-f001:**
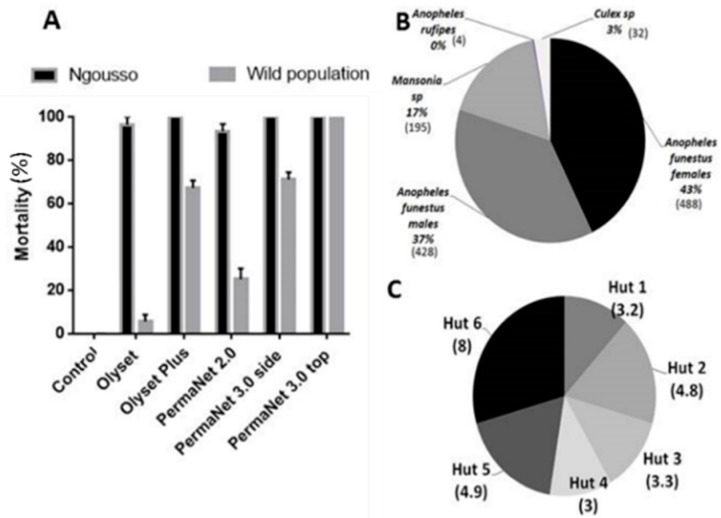
Quality control before the experimental hut trial in Mibellon: Species composition in Mibellon: (**A**) Quality control of the efficacy of all the four nets checked against the susceptible laboratory strain of *Anopheles gambiae* Ngousso and LLINs efficacy testing using cone assays against the pyrethroid resistant *An. funestus* population from Mibellon, Cameroon; (**B**) Number of mosquitoes collected during the hut effect assessment; (**C**) Average of *Anopheles funestus.* ss collected by hut during the 18 days of the hut effect investigation.

**Figure 2 genes-11-00143-f002:**
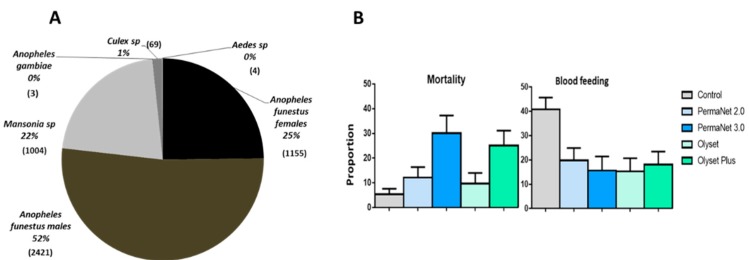
Performance of the four LLINs in experimental hut trials against pyrethroid resistant *An. funestus* in Cameroon. (**A**) Mosquito species composition during the experimental hut study. (**B**) Proportion of mortality and blood feeding rate for the four LLINs against *An. funestus.*

**Figure 3 genes-11-00143-f003:**
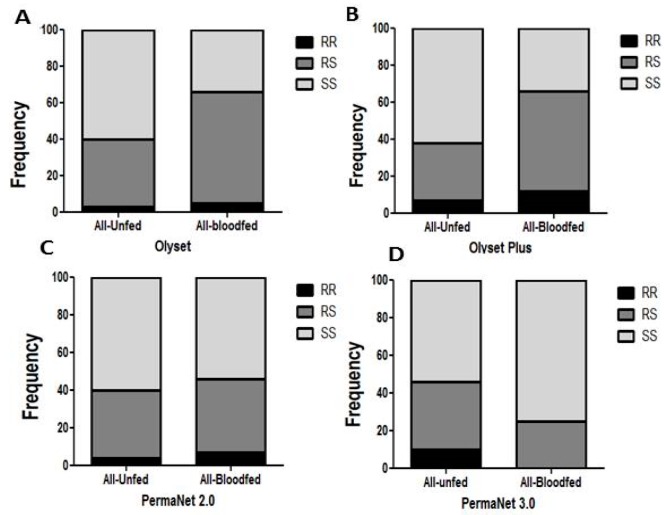
Impact of the L119F*-GSTe2* mediated metabolic resistance on bednet efficacy for blood feeding ability: (**A**) Genotype distribution of L119F-*GSTe2* between blood fed and unfed mosquitoes after exposure to Olyset showing a significant increased ability to blood feed for resistant mosquitoes; (**B**) Genotype distribution of L119F-*GSTe2* between blood fed and unfed mosquitoes after exposure to Olyset Plus showing a significant increased ability to blood feed for resistant; (**C**) Genotype distribution of L119F-*GSTe2* between blood fed and unfed mosquitoes after exposure to PermaNet 2.0 showing an inverse marginal increased ability to blood feed for homozygote susceptible SS compared to homozygote resistant RR (*p* < 0.05); (**D**) Genotype distribution of L119F-*GSTe2* between blood fed and unfed mosquitoes after exposure to PermaNet 3.0 showing an increased ability to blood feed of susceptible mosquitoes compared to resistant mosquitoes (R vs. S: OR = 0.29 *p* < 0.05).

**Figure 4 genes-11-00143-f004:**
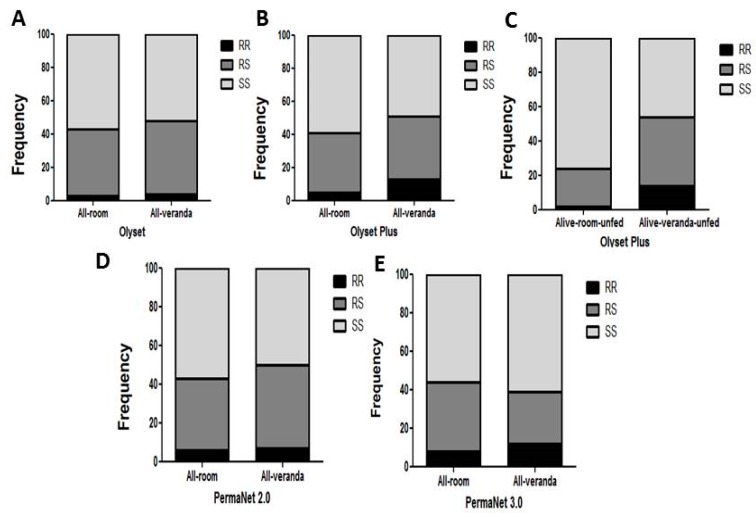
Impact of the *L119F-GSTe2* mediated metabolic resistance on bednet efficacy -exophily: (**A**) Genotype distribution of L119F-*GSTe2* between indoor (Room) and outdoor (verandah) mosquitoes after exposure to Olyset showing no association; (**B**) Genotype distribution of L119F-*GSTe2* between indoor (Room) and outdoor (verandah) mosquitoes after exposure to Olyset Plus showing a significant increased ability to exit the room when considering all mosquitoes; (**C**) A greater association is observed with exophily with Olyset Plus when only analyzing unfed mosquitoes; (**D**) Genotype distribution of L119F-*GSTe2* between indoor (Room) and outdoor (verandah) mosquitoes after exposure to PermaNet 2.0 showing a significant association (RS vs. SS: OR = 1.35; *p* < 0.01); (**E**) Genotype distribution of L119F-*GSTe2* between indoor (Room) and outdoor (verandah) mosquitoes after exposure to PermaNet 3.0 showing no association.

**Figure 5 genes-11-00143-f005:**
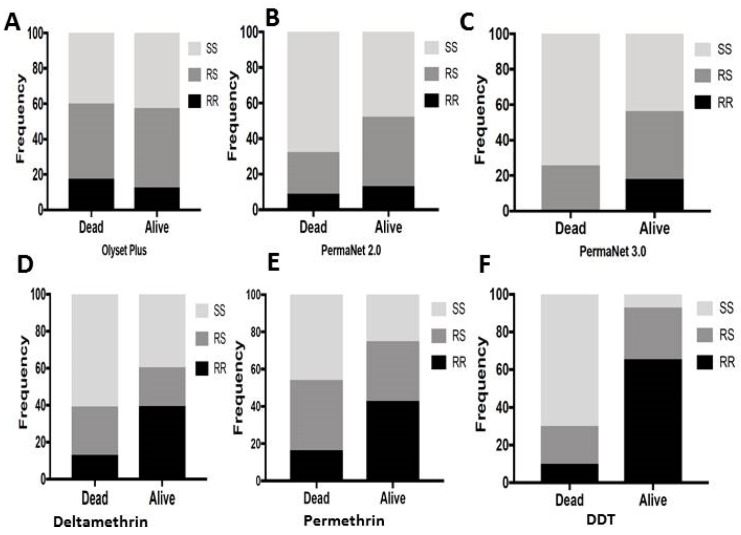
Association between L119F-*GSTe2* mutation and ability to survive exposure to LLINs (cone assays) and WHO papers (bioassays): (**A**) Genotype distribution of *L119F*-GSTe2 between alive and dead mosquitoes after exposure to Olyset Plus showing no association; (**B**) Genotype distribution of L119F-*GSTe2* between alive and dead mosquitoes after exposure to PermaNet 2.0 showing a significantly increased ability of resistant mosquitoes to survive (R vs. S: *p* < 0.01); (**C**) Genotype distribution of L119F-*GSTe2* between alive and dead mosquitoes after exposure to PermaNet 3.0 showing a significantly greater ability of resistant mosquitoes to survive than that seen for PermaNet 2.0 (RR vs. SS: OR = 30.1; *p* < 0.001); (**D**) Genotype distribution of L119F-*GSTe2* between alive (90 min) and dead (30 min) mosquitoes after exposure to deltamethrin showing a significantly greater ability of homozygote resistant mosquitoes to survive (RR vs. SS: OR = 4.6; *p* < 0.001); (**E**) Similarly for permethrin, homozygote resistant mosquitoes exhibit a significantly greater ability of to survive (RR vs. SS: OR = 4.8; *p* < 0.001); (**F**) For DDT, a much greater ability of mosquitoes with the resistance allele to survive exposure to DDT papers (RR vs. SS: OR = 66.7; *p* < 0.001).

**Table 1 genes-11-00143-t001:** Description of the long-lasting insecticidal nets used.

Treatment Arm	Description	Manufacturer
**Untreated**	100% polyester with no insecticide	Local market
**Olyset**	8.6 × 10^−4^ kg/m^2^ (2%) of permethrin incorporated into polyethylene	Sumitomo Chemical
**Olyset Plus**	8.6 × 10^−4^ kg/m^2^ (2%) of permethrin and 4.3 × 10^−4^ kg/m^2^ (1%) of Piperonyl butoxide (PBO) incorporated into polyethylene	Sumitomo Chemical
**PermaNet 2.0**	100% polyester coated with 1.8 g/kg of deltamethrin	Vestergaard Frandsen
**PermaNet 3.0**	Combination of 2.8 g/kg of deltamethrin coated on polyester with strengthened border (side panels) and deltamethrin (4.0 g/kg) and PBO (25 g/kg)	Vestergaard Frandsen

**Table 2 genes-11-00143-t002:** Results of the performance of the five brands of LLINs against wild *An. funestus* females in experimental huts.

Treatments					
	Control	PermaNet 2.0	PermaNet 3.0	Olyset	Olyset Plus
Females caught	390	237	153	176	199
Exophily%	6.9	17.7 ***	17.0 ***	29.0 ***	22.1 ***
95% Confidence limits	4.40–9.44)	(12.85–22.58)	(11.04–22.94)	(22.27–35.68)	(16.34–27.88)
Blood fed (%)	40.8	19.8 ***	15.7 ***	15.3 ***	17.2 ***
95% Confidence limits	(35.89–45.65)	(14.75–24.91)	9.92–21.45)	10.02–20.67)	(12.74–23.44)
Blood feed inh. (%)	0.0	51.36	61.52	62.37	55.63
Personal protection (%)	0.0	70.44	84.90	83.01	77.35
Overall mortality (%)	5.4	12.2 **	30.1 ***	9.7 *	25.1 ***
95% Confidence limits	(31.14–7.62)	(8.06–16.41)	(22.80–37.33)	(5.29–14.02)	(19.10–31.15)
Immediate mortality	2.6	5.4	27.6 ***	6.1 *	18.3 ***
entry rate	33.7	20.5 *	13.2 ***	15.2 ***	13.6 ***
Deterrence (%)	0.0	39.2	60.8	54.9	49.0

* = *p* < 0.05 ** = *p* < 0.01 *** = *p* < 0.001.

**Table 3 genes-11-00143-t003:** Results of the performance of the four brands of LLINs against wild *Mansonia* spp. in experimental huts.

	Control	PermaNet 2.0	PermaNet 3.0	Olyset	Olyset Plus
Total	342	142	216	141	163
Exophily%	42.4	37.3 *	31.9 **	28.4 **	31.3 **
95% Confidence limits	(37.16–47.64)	(29.37–45.28)	(25.73–38.16)	(20.93–35.81)	(24.17–38.41)
Blood feeding (%)	30.1	16.2 **	9.3 ***	15.6 ***	20.2 **
95% Confidence limits	(25.25–34.98)	(10.14–22.26)	(5.39–13.12)	(9.61–21.59)	(14.08–26.41)
Blood feeding inh. (%)	0.0	46.22	69.26	48.19	32.78
Personal protection (%)	0.0	77.66	80.53	78.64	67.96
Overall mortality (%)	37.4	51.4 **	65.5 ***	40.4 **	68.1 ***
Immediate mortality	36.8	46.5	62.0 ***	39.0 ***	65.0 ***
Entry rate	34.0	14.1 ***	21.5 **	14.0 ***	16.2 ***
Deterrence (%)	0.0	58.5	36.8	58.8	52.3

* = *p* < 0.05 ** = *p* < 0.01 *** = *p* < 0.001.

**Table 4 genes-11-00143-t004:** Impact of L119F-GSTe2 mutation on the ability of various insecticide-treated nets to kill mosquitoes.

			Mortality	
	Genotype	OR	PV	CI
**Olyset Plus**	RR vs. SS	0.61	˃0.05	0.22–1.66
	RR vs. RS	0.59	˃0.05	0.21–1.68
	RS vs. SS	1.02	1	0.56–1.8
	R vs. S	0.85	˃0.05	0.45–1.59
**PermaNet 3.0**	RR vs. SS	3.47	˃0.05	1–11.9
	RR vs. RS	3.56	˃0.05	1–12.7
	RS vs. SS	0.97	1	0.53–1.76
	R vs. S	1.42	˃0.05	0.74–2.7
**PermaNet 2.0**	RR vs. SS			
	RR vs. RS			
	RS vs. SS	1.1	1	0.6
	R vs. S	1.18	˃0.05	0.6–2.2

**Table 5 genes-11-00143-t005:** Impact of L119F-*GSTe2* mutation on the efficacy of various bed nets to prevent blood feeding.

	Genotype	OR	PV	CI
**Olyset Plus**	RR vs. SS	3.3	<0.05	1.19–9.2
	RR vs. RS	1.07	˃0.05	0.38–3.02
	RS vs. SS	3.08	<0.01	1.67–5.66
	R vs. S	2.23	<0.05	1.2–4.1
	**Mosquitoes in Room**			
	RR vs. SS	12.3	<0.001	2.5–60.4
	RR vs. RS	1.46	˃0.05	0.29–7.2
	RS vs. SS	8.42	<0.001	4.37–16.2
	R vs. S	4.56	<0.001	2.26–9.2
**Olyset**	RR vs. SS	2.6	˃0.05	0.6–11.7
	RR vs. RS	0.89	˃0.05	0.2–3.9
	RS vs. SS	2.97	<0.001	1.65–5.35
	R vs. S	2	<0.05	1.06–3.7
	**Blood feeding in Room**			
	RS vs. SS	3	<0.001	1.67–5.4
	R vs. S	1.71	˃0.05	0.89–3.3
**PermaNet 3.0**	RR vs. SS	inf		
	RR vs. RS	inf		
	RS vs. SS	0.5	<0.05	0.26–0.92
	R vs. S	0.35	<0.05	0.17–0.73
	**Blood feeding in Room**			
	RS vs. SS	0.41	˃0.05	0.21–0.77
	R vs. S	0.29	˃0.05	0.14–0.63
**PermaNet 2.0**	RR vs. SS	0.4	˃0.05	0.1–1.56
	RR vs. RS	0.48	˃0.05	0.12–1.9
	RS vs. SS	0.8	˃0.05	0.47–1.5
	R vs. S	1.31	˃0.05	0.6–2.5

**Table 6 genes-11-00143-t006:** Impact of L119F-GSTe2 mutation on the efficacy of various bed nets in repellency.

Exophily				
	Genotype	OR	PV	CI
**Olyset Plus**	RR vs. SS	3.1	˃0.05	1.1–9
	RR vs. RS	2.48	˃0.05	0.83–7.42
	RS vs. SS	1.25	˃0.05	0.67–2.26
	R vs. S	1.1	˃0.05	0.6–2.2
	**Unfed Alive**			
	RR vs. SS	11.76	<0.01	2.59–53.4
	RR vs. RS	3.9	1	0.8–18.59
	RS vs. SS	2.99	<0.01	1.58–5.6
	R vs. S	3.4	<0.001	1.67–6.9
	RS vs. SS	1.18	˃0.05	0.6–2.1
	R vs. S	1.17	˃0.05	0.62–2.2
**PermaNet 3.0**	RR vs. SS	1.22	1	0.46–3.2
	RR vs. RS	1.68	˃0.05	0.6–4.7
	RS vs. SS	0.59	˃0.05	0.31–1.12
	R vs. S	0.94	1	0.5–1.8
**PermaNet 2.0**	RS vs. SS	1.35	<0.01	0.75–2.43
	R vs. S	1.22	˃0.05	0.65–2.3
	**Unfed Alive**			
	RS vs. SS	3.37	<0.001	1.84–6.17
	R vs. S	1.8	˃0.05	0.99–3.38

**Table 7 genes-11-00143-t007:** Impact of GSTe2 on the ability of field population *An. funestus* to survive using samples from cone assays.

**Cone Assays**				
**Olyset Plus** 40 alive vs. 40 dead		**OR**	**PV**	**CI**
	RR vs. SS	1.1	˃0.05	0.4–2.7
	RS vs. SS	0.9	1.0	0.5–1.8
	RR vs. RS	1.1	˃0.05	0.4–2.7
	R vs. S	1.04	1.0	0.5–1.8
**PermaNet 2.0** 46 alive vs. 34 dead	RR vs. SS	2.09	<0.01	1.1–4.2
	RS vs. SS	2.3	<0.001	1.2–4.4
	RR vs. RS	0.8	˃0.05	0.3–2.4
	R vs. S	1.8	<0.01	0.9–3.5
**PermaNet 3.0** 39 alive vs. 31 dead	RR vs. SS	30.1	<0.001	0.8–5.3
	RS vs. SS	2.4	<0.01	1.3–4.5
	RR vs. RS	12.3	<0.001	1.5–98.6
	R vs. S	3.8	<0.001	1.8–7.7
**WHO Bioassays**				
**Deltamethrin** 38 alive vs. 38 dead		**OR**	**PV**	**CI**
	RR vs. SS	4.6	<0.001	2.5–8.4
	RS vs. SS	1.22	˃0.05	0.69–2.15
	RR vs. RS	3.75	<0.001	2.1–6.8
	R vs. S	2.80	<0.01	1.54–5.1
**Permethrin** 28 alive vs. 37 dead	RR vs. SS	4.8	<0.001	2.66–8.8
	RS vs. SS	1.6	˃0.05	0.86–2.8
	RR vs. RS	3.1	<0.001	1.7–5.5
	R vs. S	2.99	<0.001	1.7–5.3
**DDT** 29 alive vs. 30 dead	RR vs. SS	66.7	<0.001	27.1–164
	RS vs. SS	14.0	<0.001	6.3–31.2
	RR vs. RS	4.8	<0.001	2.4–9.5
	R v S	15.3	<0.001	7.7–30.5

OR, Odds ratio; PV, *p* value; CI, Confidence interval.

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
