# Peer review of "An Experimental Hut Evaluation of PBO-Based and Pyrethroid-Only Nets against the Malaria Vector Anopheles funestus Reveals a Loss of Bed Nets Efficacy Associated with GSTe2 Metabolic Resistance"

_genes, 2020, doi:10.3390/genes11020143_

Round 1

Reviewer 1 Report

The impact of insecticide resistance on LLIN is efficacy of great concern and debate in the vector control community. The study is timely and welcomed as the effect of other resistance mechanisms aside from P450-based and kdr on the entomological efficacy of LLIN is over looked. The authors have used entomological outcomes as a proxy for malaria cases. The following comments warrant consideration;

Key malaria epidemiology figures would place the study in context for readers and it is also important to state that progress in malaria control has stalled and reference the World Malaria Report 2018 Linked to the above it is also important to highlight that there are other factors challenging malaria control in addition to insecticide resistance in mosquitoes; e.g. reduced donor funding and drug resistance. Please provide a statement for WHO’s 2017 recommendation for the deployment of PBO nets (ie. National malaria control programmes and their partners should consider the deployment of pyrethroid-PBO nets in areas where the main malaria vector(s) have pyrethroid resistance that is: a) confirmed, b) of intermediate level (as defined above), and c) conferred (at least in part) by a monooxygenase-based resistance mechanism, as determined by standard procedures.) It is important for the reader that they understand that the study is looking at LLIN efficacy regarding entomological outcomes not malaria incidence or prevalence therefore lines 50-51 should be more explicit about this The lack of insight into the selection of and selective reporting of results for Yorkool is confusing. The authors do not provide a rationale for its inclusion. Results for the cone assay are presented for Yorkool as are those for the experimental hut studies but there are no results for this net exploring the impact of L119F-GSTe2. Furthermore, Yorkool does not feature in the discussion at all. The authors should either; provide a rationale for the inclusion of this net, present all associated results (if not explain why the impact of L119F-GSTe2 was not tested for this net) and include it in the discussion or exclude Yorkool entirely from the manuscript. As per WHO protocol the study looked at delayed mortality (i.e. 24hrs) following net exposure. I would like the authors to place their findings in the context of a study by Viana et al (2016) who discuss the implications of looking at mortality beyond 24 hrs in resistant mosquitoes and the implication for LLIN efficacy. Correct me if wrong but 100% mortality in the cone assay with the top of PN 3.0 which contains PBO indicates that L119F-GSTe2 is not affecting efficacy of this net, only the side of the net minus PBO. This result would suggest that rather than add a GST inhibitor as the authors suggest incorporating PBO over the entire net may be a better approach. Please address this is the discussion. In addition to point 7 there was no correlation between mosquito mortality and L119F-GSTe2 for PN 3.0, PN 2.0 and Olyset Plus. If resistant mosquitoes with L119F-GSTe2 are still being killed the nets can still reduce the local population with fewer mosquitoes available to feed even if blood feeding is associated with resistance. The abstract is therefore misleading as it gives the impression that PBO nets could be jeopardised, but in terms of mosquito mortality the results do not point to this conclusion. The improved efficacy of PN 3.0 compared PN 2.0 is attributed to the higher dose of deltamethrin (line 548). On the same theme the paper would benefit from a discussion in context of the results presented of the increased exposure of permethrin on the surface of Olyset Plus compared with Olyset as discussed by Sovmand (2018) The authors suggest that the addition of diethyl maleate on LLIN could mitigate the effect of GST-based activity. This is a bold statement to make and could have been better supported if the authors had performed tests pre-exposing An. funestus to a GST inhibitor followed by LLINs using the cone assays to gauge if an effect could be observed. To reiterate point 7 the results suggest a version of PN 3.0 with PBO over the entire net could be an option. Following on from the suggestion incorporating a GST inhibitor to LLIN, based on the findings of the study do the authors recommend that this would be best served on deltamethrin nets or should this include permethrin nets? Do the results indicate whether an inhibitor would be best served at the top of the net (e.g. PN 3.0) or over the entire net (e.g. Olyset plus)? The following edits should be made to the methods section; Please provide some information on malaria epidemiology for Mibellon State how many nights the EHT ran for Confirm whether huts were cleaned each morning to avoid collector bias Confirm whether the collectors were blinded to the treatment type in each hut to avoid bias Personal protection presented in the results but how it is calculated is not described in the methods so please state how it was done    There are several errors in the presentation of the results; Fig1A: Mortality should be % mortality Line 240: should read Fig. 1C not 1D Fig 2B and 2C are presented the wrong way around in the figure legend Table 2: to make table 2 readable present the P values as asterisks (i.e. p<0.05 = *, <0.01 =** and <0.001 = ***) There are number of formatting issues with results tables with brands of LLIN not aligned in some cases

Author Response

Reviewer 1 comments

Reviewer:

Key malaria epidemiology figures would place the study in context for readers and it is also important to state that progress in malaria control has stalled and reference the World Malaria Report 2018 Linked to the above it is also important to highlight that there are other factors challenging malaria control in addition to insecticide resistance in mosquitoes; e.g. reduced donor funding and drug resistance.

Response: We thank the reviewer for this suggestion. This information has now been added in the introduction section of the manuscript as suggested.

Reviewer: Please provide a statement for WHO’s 2017 recommendation for the deployment of PBO nets (ie. National malaria control programmes and their partners should consider the deployment of pyrethroid-PBO nets in areas where the main malaria vector(s) have pyrethroid resistance that is: a) confirmed, b) of intermediate level (as defined above), and c) conferred (at least in part) by a monooxygenase-based resistance mechanism, as determined by standard procedures.)

Response: This information has been added in the introduction as suggested.

Reviewer: It is important for the reader that they understand that the study is looking at LLIN efficacy regarding entomological outcomes not malaria incidence or prevalence therefore lines 50-51 should be more explicit about this

Response: A statement has been added in the related sentence in introduction to further specify that this impact is related to the entomological outcomes.

Reviewer: The lack of insight into the selection of and selective reporting of results for Yorkool is confusing. The authors do not provide a rationale for its inclusion. Results for the cone assay are presented for Yorkool as are those for the experimental hut studies but there are no results for this net exploring the impact of L119F-GSTe2. Furthermore, Yorkool does not feature in the discussion at all. The authors should either; provide a rationale for the inclusion of this net, present all associated results (if not explain why the impact of L119F-GSTe2 was not tested for this net) and include it in the discussion or exclude Yorkool entirely from the manuscript.. 

Response: As suggested by the reviewer, we have now removed the Yorkool net from the study and the tables changed accordingly.

Reviewer: As per WHO protocol the study looked at delayed mortality (i.e. 24hrs) following net exposure. I would like the authors to place their findings in the context of a study by Viana et al (2016) who discuss the implications of looking at mortality beyond 24 hrs in resistant mosquitoes and the implication for LLIN efficacy.

Response: In our study we did not observe the delay mortality beyond 24 hours, so it will be difficult to presume what could have happened. However, we agree with the reviewer that this issue important and we will take this into account in future study.

Reviewer: Correct me if wrong but 100% mortality in the cone assay with the top of PN 3.0 which contains PBO indicates that L119F-GSTe2 is not affecting efficacy of this net, only the side of the net minus PBO. This result would suggest that rather than add a GST inhibitor as the authors suggest incorporating PBO over the entire net may be a better approach. Please address this in the discussion.

Response: We have taken into account the comment made by the reviewer in Discussion by adding the following sentence at the end: “Nevertheless the greater efficacy seen with the roof of PermaNet 3.0 in cone bioassays shows that P450 role remains important.  Future studies will need to also perform synergist assays with DEM and PBO to establish the contribution of both GSTs and cytochrome P450s to better inform of the likely efficacy of LLINs.”

Reviewer: In addition to point 7 there was no correlation between mosquito mortality and L119F-GSTe2 for PN 3.0, PN 2.0 and Olyset Plus. If resistant mosquitoes with L119F-GSTe2 are still being killed the nets can still reduce the local population with fewer mosquitoes available to feed even if blood feeding is associated with resistance. The abstract is therefore misleading as it gives the impression that PBO nets could be jeopardised, but in terms of mosquito mortality the results do not point to this conclusion.

Response: We have taken the reviewer’s comment into account and modified the related sentence in abstract by removing “jeopardising” and stating that this impact concerns blood feeding inhibition. The sentence now reads as follows: “This study shows that the  efficacy of PBO-based nets against pyrethroid resistant malaria vectors (e.g. blood feeding inhibition) could be impacted by other mechanisms including GST-mediated metabolic resistance, not affected by the synergistic action of PBO”

Reviewer: The improved efficacy of PN 3.0 compared PN 2.0 is attributed to the higher dose of deltamethrin (line 548). On the same theme the paper would benefit from a discussion in context of the results presented of the increased exposure of permethrin on the surface of Olyset Plus compared with Olyset as discussed by Sovmand (2018) The authors suggest that the addition of diethyl maleate on LLIN could mitigate the effect of GST-based activity. This is a bold statement to make and could have been better supported if the authors had performed tests pre-exposing An. Funestus to a GST inhibitor followed by LLINs using the cone assays to gauge if an effect could be observed. To reiterate point 7 the results suggest a version of PN 3.0 with PBO over the entire net could be an option. Following on from the suggestion incorporating a GST inhibitor to LLIN, based on the findings of the study do the authors recommend that this would be best served on deltamethrin nets or should this include permethrin nets? Do the results indicate whether an inhibitor would be best served at the top of the net (e.g. PN 3.0) or over the entire net (e.g. Olyset plus)?

Response: We thank the reviewer for these useful comments. Regarding the paper by Svokmand (2018) claiming a difference in % of active ingredient between Olyset and Olyset plus, we don’t believe that this is the case as both nets have 2% of permethrin as we further confirmed on the WHO prequalification listing. That is why we only discussed this difference for PrmaNet 3.0 (4% deltamethrin) and PermaNet 2.0 (2%). We agree that it would have been better to also perform a DEM synergist assay to further validate role of GST but the correlation we observed between L119F-GSTe2 and mortality after both bioassays and cone assays is a strong indication that GSTs play a role. Nevertheless, we should perform DEM synergist assays in future studies. At this stage, we also cannot say for sure whether addition of DEM will act better with permethrin or deltamethrin nets since many factors may impact this including concentration of pyrethroids and background resistance notably level of cytochrome P450. It is better to say that this interesting question needs to be further assessed in future study same as whether DEM could be better suited on the top or on the whole net.  We have now added the following sentence at end of conclusion: “However, further studies will need to be performed before including establishing the extend of synergism provided by DEM in various mosquito populations and the most suitable position on the net (either on top as PBO or on the entire net).”

Reviewer: Please provide some information on malaria epidemiology for Mibellon.

Response: We currently do not have the parasitological data on this site for humans. However, we have provided further information with the following sentence in Methods: “Malaria transmission is perennial with a high transmission as shown by very high infection rate of Plasmodium infection caused by Plasmodium falciparum but also P. malariae [1] (Tchouakui et al 2019). “

Reviewer: State how many nights the EHT ran for Confirm whether huts were cleaned each morning to avoid collector bias Confirm whether the collectors were blinded to the treatment type in each hut to avoid bias

Response: All these details were added in the methods section from line 151

Reviewer: Personal protection presented in the results but how it is calculated is not described in the methods so please state how it was done.

Response: It is mentioned from line 185 Personal protection (%) = 100 x (Bu – Bt) / Bu, where Bu is the total number of blood-fed mosquitoes in the huts with untreated nets and Bt is the total number of blood-fed mosquitoes in the huts with treated nets

Reviewer:  There are several errors in the presentation of the results; Fig1A: Mortality should be % mortality Line 240: should read Fig. 1C not 1D Fig 2B and 2C are presented the wrong way around in the figure legend Table 2: to make table 2 readable present the P values as asterisks (i.e. p<0.05 = *, <0.01 =** and <0.001 = ***) There are number of formatting issues with results tables with brands of LLIN not aligned in some cases 

Response: Figure 1 and Figure 2 were modified. 1D was modified to 1C see line 264. Table 1, 2 and 3 were adjusted as requested.

Reviewer 2 Report

This is a robust study, pending changes to certain caveats of this study. The study assumes that mosquitoes collected from experimental huts only posses L to F mutation in the GSTe2 gene, which is a flawed approach. Many publications report P450-based and KDR-mediated pyrethroid resistance in malaria vector Anopheles funestus. In light of those reports, how can the authors say that LLIN/pyrethroid-resistance is linked solely with GST-mediated metabolic resistance? To address this caveat, authors can include the words “is partially linked” in the manuscript title and also include clarification of this point throughout the manuscript. Since the authors did not screen hut-collected mosquitoes for overexpression of P450-associated genes and KDR mutation status, it is imperative to discuss the potential contribution of P450 and KDR resistance mechanisms to pyrethroid resistance in RR, RS and SS genotypes for the L to F GSTe2 mutation. One of the reasons for the lack of significant correlation between L to F mutation and many entomological outcomes (except blood feeding, exophily and mortality in cone bioassays) could be because of P450 and KDR mechanisms in mosquitoes used in this study. Thus, overall it is important to discuss the relative contribution of P450 and KDR mechanisms to the resistant phenotype.

I think the length of the manuscript is too long. Authors can condense the information by reediting the manuscript for conciseness and use of correct grammar. There are many grammatical, sentence construction and formatting errors that need to be addressed. I have pointed out some of the grammatical and formatting errors in my specific comments below, but there are likely many more present. Another suggestion for reducing the length of this manuscript would be to present data/ figures for entomological outcomes that did not show significant association with GSTe2 mutation in the supplementary information section. This will greatly reduce the length of the manuscript and make it more focused/ to the point.

Specific comments:

Lines 28-31: Detailed discussion on the differences between permethrin and deltamethrin treated nets is warranted in the discussion section. Perhaps GSTe2 is more involved in phase II metabolism of permethrin and not deltamethrin.

Lines 36-37: Authors can include a brief discussion on the topic of DEM incorporated bed nets in the discussion section.

Line 53: Change "the" to "this'

Lines 54-57: Rephrase this whole sentence as it is not grammatically correct and also too long. Consider splitting this sentence into two short easy to understand and grammatically correct sentences.

Line 77: Delete "the" after "provides"

Lines 79 to 83: Authors have included information on treatments that they compared in the introduction section, which is not a conventional way of writing the last paragraph of the introduction section. Instead, I recommend the authors to clearly state the hypothesis and/or objectives of this study and move the first sentence on product comparisons to the Materials and Methods section. Perhaps there is already a sentence like this in the Methods section.

Line 84: Whenever appropriate Materials and Methods section should be written in past tense.

Line 86-87: Change "is" to "was" in line 86 for correct use of tense.

Sub-section 2.4; Line 133: Although you are following the WHO hut testing protocol, it will be important to explicitly state how many replicates were conducted and what was the duration of each replicate. This information is not clearly identifiable in the methods section.

Line 181: Add "the efficacy" after "on" in the heading for sub-section 2.5.

Line 200: In the "data analysis" section please mention the probability level or p-value cut-off that was used for different statistical tests.

Line 269: Change to "significantly"

Line 286: Change to "significantly"

Line 319: There are some heading formatting issues that need to be corrected in this sub-section.

Line 325: Odd Ratio is not abbreviated as OR in the methods/ data analysis section. It is important to mention the abbreviation in parenthesis when the words are spelled out completely for the first time.

Line 326: Since this is a non-entomology journal it might be beneficial to spell out these (RR, RS and SS) abbreviations at their first mention in the text or the table.

Line 377: Something is not grammatically correct with this sentence.

Line 469: Change "have" to "has"

Line 491: The term excito-repellency has not been mentioned in the introduction or results section. So how do you directly start talking about it in the discussion section. There has to be some connection between what is mentioned in the results and in the discussion sections.

Author Response

Reviewer 2 comments

Reviewer: This is a robust study, pending changes to certain caveats of this study. The study assumes that mosquitoes collected from experimental huts only posses L to F mutation in the GSTe2 gene, which is a flawed approach. Many publications report P450-based and KDR-mediated pyrethroid resistance in malaria vector Anopheles funestus. In light of those reports, how can the authors say that LLIN/pyrethroid-resistance is linked solely with GST-mediated metabolic resistance? To address this caveat, authors can include the words “is partially linked” in the manuscript title and also include clarification of this point throughout the manuscript

Response: Thanks you for this comment The title has been modified

Reviewer: Since the authors did not screen hut-collected mosquitoes for overexpression of P450-associated genes and KDR mutation status, it is imperative to discuss the potential contribution of P450 and KDR resistance mechanisms to pyrethroid resistance in RR, RS and SS genotypes for the L to F GSTe2 mutation. One of the reasons for the lack of significant correlation between L to F mutation and many entomological outcomes (except blood feeding, exophily and mortality in cone bioassays) could be because of P450 and KDR mechanisms in mosquitoes used in this study. Thus, overall it is important to discuss the relative contribution of P450 and KDR mechanisms to the resistant phenotype.

Response: We agree with the reviewer that other mechanisms are at play in Mebellon than just GST as shown by a recent publication of Weedall et al (2019) highlighting the over-expression of cytochrome P450 genes such as CYP6P5 and CYP325a. However, there is not yet a molecular marker to help assess the impact of these markers on the different samples we could after experimental huts as some are dead and could not be used for RNA extraction. Nevertheless, we have now added in the discussion the potential role of P450s. For the Kdr resistance, there is no Kdr resistance detected yet in Anopheles funestus (Irving et Wondji 2017) as confirmed also in Mibellon (Menze et al 2018).  This is now added in Discussion at end of section on Experimental huts and GSTe2.

Reviewer: Another suggestion for reducing the length of this manuscript would be to present data/ figures for entomological outcomes that did not show significant association with GSTe2 mutation in the supplementary information section. This will greatly reduce the length of the manuscript and make it more focused/ to the point.

Response: Some figures like Fig 1 have been modified and figure 3 on mortality has been moved to supplementary files as suggested.

Reviewer: Lines 28-31: Detailed discussion on the differences between permethrin and deltamethrin treated nets is warranted in the discussion section. Perhaps GSTe2 is more involved in phase II metabolism of permethrin and not deltamethrin.

Response:

We have now added the following sentence in Discussion: “Functional analyses with transgenic Drosophila, In vitro metabolism assays with recombinant GSTe2 enzyme combined with genotype/phenotypes analyses had shown that GSTe2 was able to confer resistance to both permethrin and deltamethrin but more so to permethrin (Riveron et al 2014; Riveron et al 2017). This could partly explain the differences observed between permethrin- and deltamethrin-based nets although further studies will be needed to fully establish the underlying reason.”

Reviewer Lines 36-37: Authors can include a brief discussion on the topic of DEM incorporated bed nets in the discussion section.

Response: This has now been done in the last section of the Discussion

Reviewer: Line 53: Change "the" to "this'

Response: This has been done, see line 67

Reviewer: Lines 54-57: Rephrase this whole sentence as it is not grammatically correct and also too long. Consider splitting this sentence into two short easy to understand and grammatically correct sentences.

Response: The sentence has been rephrased and split in two as follow: . Recent efforts have detected a key genetic marker in the glutathione S-transferase epsilon 2 gene (GSTe2). This marker confers metabolic-mediated resistance to pyrethroids and Dichlorodiphenyltrichloroethane (DDT) in the major malaria vector Anopheles funestus in West and Central Africa. See from line 67

Reviewer: Line 77: Delete "the" after "provides"

Response: This has been done. See from Line 92

Reviewer: Lines 79 to 83: Authors have included information on treatments that they compared in the introduction section, which is not a conventional way of writing the last paragraph of the introduction section. Instead, I recommend the authors to clearly state the hypothesis and/or objectives of this study and move the first sentence on product comparisons to the Materials and Methods section. Perhaps there is already a sentence like this in the Methods section.

Response: This has been done. See Line 94 and 95

Reviewer: Line 84: Whenever appropriate Materials and Methods section should be written in past tense.

Response: We have now checked this and use past tense whenever possible.

Reviewer Line 86-87: Change "is" to "was" in line 86 for correct use of tense.

Response: This has been done. See from Line 101

Reviewer: Sub-section 2.4; Line 133: Although you are following the WHO hut testing protocol, it will be important to explicitly state how many replicates were conducted and what was the duration of each replicate. This information is not clearly identifiable in the methods section.

Response: information about the duration of each test is mentioned in line146 and 147. And the number of replicate and the number per replicate was mentioned in line 148.

Reviewer Line 200: In the "data analysis" section please mention the probability level or p-value cut-off that was used for different statistical tests.

Response: Thank you for this observation. In this study, for all the analyses, an alpha of 0.05 was used as the cut off for significance. This was added at the end of  the Data analysis section.

Reviewer Line 269: Change to "significantly"

Line 286: Change to "significantly"

Response:  Thank you for the remark, it has been modified

Reviewer: Line 319: There are some heading formatting issues that need to be corrected in this sub-section.

Response: This has been corrected line 353

Reviewer: Line 325: Odd Ratio is not abbreviated as OR in the methods/ data analysis section. It is important to mention the abbreviation in parenthesis when the words are spelled out completely for the first time.

Response: This  has  now been done (see line 234)

Reviewer: - Line 326: Since this is a non-entomology journal it might be beneficial to spell out these (RR, RS and SS) abbreviations at their first mention in the text or the table.

Response: This has been done in Line 210

Discussion

Reviewer: Line 377: Something is not grammatically correct with this sentence

Response: This has been corrected in Line 404

Reviewer: Line 469: Change "have" to "has"

Response: This has been done in Line 504

Reviewer: Line 491: The term excito-repellency has not been mentioned in the introduction or results section. So how do you directly start talking about it in the discussion section. There has to be some connection between what is mentioned in the results and in the discussion sections.

Response:  We agree with the reviewer that this term should have been mentioned before. This has now been mentioned in both methods and Results stating that it is similar to exophily.

Round 2

Reviewer 1 Report

I thank the authors for the considered response to my comments and feel the manuscript is much improved. There a few minor amendments that need to be addressed;

Tables 2 & 3 provide a footnote stating the level of significance for * ie 0.05, ** 0.01 and *** 0.001 Line 582 put gene names in italics Line 620: 'Is reason why as suggested by WHO 2017 report we recommend the deployment of PBO nets in area where pyrethroid 621 resistance is confirmed and at least partly conferred by P450' should be corrected to 'It is the reason why as suggested by WHO 2017 report we recommend the deployment of PBO nets in area where pyrethroid resistance is confirmed and at least partly conferred by P450'

Author Response

Reviewer: Tables 2 & 3 provide a footnote stating the level of significance for * ie 0.05, ** 0.01 and *** 0.001

Response: We thank the reviewer for this suggestion. A foot note stating the level of significance has been provided for table 2 and 3 as follow: *= P<0,05  **= P<0,01  ***= P<0,001

Reviewer: Line 582 put gene names in italics

Response: We thank the reviewer for this remark. The genes names have been put in italics in lines 582 and all over the manuscript.

Reviewer: Line 620: 'Is reason why as suggested by WHO 2017 report we recommend the deployment of PBO nets in area where pyrethroid 621 resistance is confirmed and at least partly conferred by P450' should be corrected to 'It is the reason why as suggested by WHO 2017 report we recommend the deployment of PBO nets in area where pyrethroid resistance is confirmed and at least partly conferred by P450'

Response: We thank the reviewer for this correction. The sentence has been corrected as suggested. See line 620.